# A Comparative Multicenter Cohort Study Evaluating the Long-Term Influence of the Strict Lockdown during the First COVID-19 Wave on Lung Cancer Patients (ARTEMISIA Trial)

**DOI:** 10.3390/cancers15245729

**Published:** 2023-12-06

**Authors:** Olivier Molinier, Camille Guguen, Marie Marcq, Anne-Laure Chene, Philippe Masson, Frédéric Bigot, Fabrice Denis, Fabienne Empereur, Philippe Saulnier, Thierry Urban

**Affiliations:** 1Respiratory Medicine Department, Hospital Center, 194 Avenue Rubillard, CEDEX 9, 72037 Le Mans, France; 2Respiratory Medicine Department, Hospital Center, 85925 La Roche-sur-Yon, France; 3Respiratory Medicine Department, Thorax Institute, University Hospital Center, University of Nantes, 44093 Nantes, France; 4Respiratory Medicine Department, Hospital Center, 49300 Cholet, France; 5Oncology Department, Western Cancer Institute Paul Papin, 49105 Angers, France; frederic.bigot@ico.unicancer.fr; 6Oncology Department, Clinique Victor Hugo, 72000 Le Mans, France; f.denis@ilcgroupe.fr; 7Regional Cancer Network ONCOPL, 44093 Nantes, France; 8Biostatistics Department, National Institute of Health and Medical Research, University of Angers, 49100 Angers, France; 9Respiratory Medicine Department, University Hospital Center, 49100 Angers, France; thurban@chu-angers.fr

**Keywords:** COVID-19, epidemiology, real-world, quality of healthcare, lung neoplasms

## Abstract

**Simple Summary:**

The consequences of the first lockdown and the accompanying strict health restrictions on lung cancer patients during the first COVID-19 wave are a public health issue unresolved to date, due to limited clinical data and the lack of extended follow-up in published studies, most of which are registry-based. The ARTEMISIA trial is an exposed–unexposed study (one cohort performed in 2019, the other in 2020), which relied on a systematic medical review of over 800 patients and suggested that the health restrictions did not impact the outcome of lung cancer patients. Paradoxically, care delays after the first imaging were shortened during the COVID-19 period and, in multivariate analysis, the prognosis of the 2020 population was improved, probably linked to the extension of indications for immunotherapy and targeted therapies. To our knowledge, this study is the only one published with a median follow-up for overall survival exceeding 2 years, and, more broadly, this trial illustrates the adaptability of healthcare structures in a strict lockdown context.

**Abstract:**

The consequences of the strict health restrictions during the first wave of COVID-19 on lung cancer (LC) patients are not known. This cohort study evaluated the impact of the initial lockdown on management of and long-term outcome in LC patients. This exposed–unexposed-type study included two evaluation periods of 6 months each in non-selected patients; one began on the first day of lockdown in 2020, and the other in 2019 during the same calendar period. Various indicators were compared: clinical profiles, management delays and overall survival beyond 2 years. A total of 816 patients from 7 public or private centers were enrolled. The clinical characteristics of the patients in 2020 did not differ from those in 2019, except that the population was older (*p* = 0.002) with more non-smokers (*p* = 0.006). Delays for pre-therapeutic medical management were generally reduced after the first imaging in 2020 (1.28 [1.1–1.49]). In the multivariate analysis, being part of the 2020 cohort was correlated with better prognosis (HR = 0.71 [0.5–0.84], *p* < 0.001). The gain observed in 2020 mainly benefited non-smoking patients, along with ECOG PS 0–2 (*p* = 0.01), stage 4 (*p* = 0.003), squamous cell carcinoma (*p* = 0.03) and receiving systemic therapy (*p* = 0.03). In conclusion, the first lockdown did not exert any deleterious impact on LC patients.

## 1. Introduction

When the COVID-19 pandemic spread to Europe in early 2020, people presenting COVID-19 symptoms were advised to stay home in the absence of a serious clinical picture to preserve an overloaded medical system. These recommendations impacted the management of cancer patients in various ways, particularly in the pre-therapy phase. The disruptions caused by the COVID-19 pandemic and the first lockdown led to a substantial decrease in new cancer cases [1,2,3] and resulted in increased diagnostic delays [4,5] and pathologic upstaging [3,6]. Certain subpopulations were particularly susceptible to diagnostic delays, such as those in early stages, women [4] and elderly individuals [7]. The long-term consequences have so far been the subject of few studies, generally concluding there was a detrimental effect of this pandemic on the prognosis of cancer patients [8,9,10].

In France, the diagnostic and therapeutic management of lung cancer (LC) patients is provided mainly (1) by qualified hospital pulmonology departments or (2) in general oncology centers or private centers. Since the start of the pandemic, most pulmonology departments have been entrusted with the treatment of COVID-19 patients. As a result, new priorities had to be defined urgently, possibly to the detriment of LC patients. However, other oncology centers, which did not have to accommodate COVID-19 patients, were paradoxically able to take care of LC patients, sometimes at an early stage, especially when consultation slots or day-hospital sessions were freed up. In parallel, the strict lockdown imposed on the entire population as of 17 March 2020 was accompanied by restrictive measures in all healthcare establishments, particularly in imaging sectors, laboratories and operating rooms, where activity was significantly reduced or even interrupted. All these measures may have contributed to a delay in the diagnosis and the treatment of patients with LC-related symptoms. On 18 March 2020, a group of experts from the Oncology Group of the French Society for Respiratory Medicine issued a document containing recommendations for reducing the number of non-urgent procedures to limit patient exposure to the virus [11]. These recommendations led to modified treatment strategy in a third of cases, especially for immune checkpoint inhibitor administration [12]. In the same period, clinical practice recommendations for the management of LC patients based on another expert consensus were published [13,14,15,16,17].

The aim of this retrospective multicenter study was to assess the immediate consequences of the pandemic and the first lockdown regarding the clinical presentation and treatment delays in LC patients, as well as its long-term prognostic impact, by comparing two cohorts of patients, one exposed to COVID-19 and another unexposed.

## 2. Materials and Methods

### 2.1. Research Design

This study was multicenter, observational, regional, retrospective and of the exposed/unexposed type. It included two cohorts: the first one (2020 cohort) consisted of patients diagnosed with LC during the 2 months of lockdown and the following 4 months, between 17 March and 28 August 2020; the second cohort (2019 cohort) covered the same calendar period but outside of any epidemic context.

The data from all patients monitored during these two periods were collected from the centralized cancer file service (CCFS), which is a shared and secure regional digital record promoted by the regional cancer network Onco-Pays de la Loire (Onco PL). Therapeutic decisions made during the multidisciplinary cancer conferences (MCCs) were systematically recorded there for each new case, regardless of the clinical presentation and any decision to withhold treatments. All centers in the PL region equipped with a thoracic oncology clinical research team and submitting each case to the CCFS were contacted. This selection was independent of the center’s operating mode, whether it was a university or general hospital, Comprehensive Cancer Center (CCC) or private center. The study was sponsored by the Centre Hospitalier du Mans, with logistical support from the University of Angers for data management and access to computer tools and the regional cancer network Onco PL for access to the CCFS.

The data collection procedure was as follows: all cases presented at the MCCs during the two defined periods were selected by a clinical study technician from the sponsoring center, who extracted anonymized clinical and therapeutic data from the CCFS. After eligibility verification, these details were transcribed into the electronic Case Report Form (eCRF) provided by the Ennov Clinical digital platform, which offered secure applications for entering clinical data in healthcare centers. The principal investigators of the study regularly convened to evaluate the consistency and exhaustiveness of the entered data. After several updates, survival data were censored as of 31 March 2023.

At a regulatory level, the study obtained approval from the French Data Protection Commission (FDPC) on 18 September 2020. In accordance with French legislation, an information letter was provided to living patients, and only those who did not object could be included. Patient anonymity was consistently preserved: the identities of patients listed in the CCFS were initially accessible in a secure manner only to the physicians responsible for their management or the clinical research technician affiliated to the sponsoring center, following FDPC approval. Similarly, the Ennov Clinical data hosting tool did not display full names or complete birthdates and each data transmission between centers was secure.

### 2.2. Inclusion/Exclusion Criteria and Collected Criteria

All patients treated in participating centers and presenting at MCCs between 17 March and 28 August of the two relevant years were eligible. They needed to be over 18 years old and their diagnosis had to be histologically confirmed regardless of the stage, histology or performance status (PS). The diagnostic date coincided with the date of the officially signed histological report. Exclusion criteria were as follows: absence of LC diagnosis; patients followed in a non-participating center; patient’s refusal to participate in the study; patient deemed incapable of providing consent; patient deprived of freedom for legal reasons or under guardianship. The collected criteria were demographic (age, sex, smoking status, type of treating centers), clinical (ECOG [Eastern Cooperative Oncology Group] PS 0–1, 2, 3–4), Stage (I–II, III, IV), histological type, diagnostic method, specialist type and molecular (genomic alterations). Regarding treatment, it could involve surgery, radiochemotherapy, radiotherapy for isolated lung lesions with ablative intent or systemic treatment, including chemotherapy alone, chemoimmunotherapy, immunotherapy or targeted therapy. Any patient not receiving one of these treatments was considered to be receiving exclusive supportive care. Palliative treatments, whether surgery, radiotherapy or other, were not reported. Simultaneously, multiple dates needed to be collected: date of first symptoms, date of the first imaging suggesting cancer (whether metastasis or primary tumor), date of the first specialist consultation, date of diagnosis (date of signed anatomopathological report), date of MCC, date of first treatment, date of progression and date of death. These dates were used to define treatment delays, as defined in Figure 1. Delays 3, 4, 5 and 6 are consistent with those published by the National Cancer Institute (https://www.e-cancer.fr/content/download/63219/569061/file/Delais-prise-en-charge-quatre-cancers-plus-frequents-V2.pdf (accessed on 1 June 2013), to which delays 1, 2 and 7 have been added. Two delays should be highlighted: delay 1 (or access to imaging) from first symptoms to first imaging, a period during which the disease was undetected although patients were symptomatic, and delay 6 (or medical management), encompassing the journey from disease suspicion to treatment, including key dates in the treatment process, such as the first consultation, diagnosis date, date of MCC and date of the first treatment.

### 2.3. Statistical Methods

Categorical variables are expressed as frequencies and percentages, while continuous variables are presented as means with standard deviations and medians with interquartile ranges (p25–p75). Comparisons of characteristics between the two cohorts were conducted using the Welch test for quantitative variables and the Pearson’s Chi-squared test (or Fisher’s exact test when applicable) for qualitative variables, with a bilateral alpha level set at 5%.

Median delays are reported either in days for treatment periods or in months for survival data, along with their 95% confidence intervals. Overall survival was defined as the time between the diagnosis date and the date of death, regardless of the cause. Progression-free survival, applicable only to patients who received treatment, was defined as the time between the diagnosis date and either the date of disease progression or death. Living patients at the end of the study were censored at the date of their last follow-up. The final data extraction was set on 31 March 2023. Regarding the comparison among treatment delays in both the overall population and subpopulations, cumulative event probabilities were estimated using the Kaplan–Meier method. The Cox Proportional Hazards model was employed for hazard ratio (HR) estimation, along with a 95% confidence interval and *p*-value. The same methodology was used for comparing survival data between the two cohorts, including both overall results and subgroup analyses.

In order to determine independent prognostic factors, a Cox Proportional Hazards analysis was applied to the entire study population. The variables included in the model were those previously identified as prognostic factors in LC (age, sex, histology, stage, smoking status, ECOG PS), and the type of cohort was added. Mutation status was not considered since it was not systematically conducted across the entire population (limited to non-squamous non-small cell carcinomas). Treatment was also not included in the model, as it depended on the initial patient characteristics. The results are presented as HR with corresponding *p*-values.

## 3. Results

### 3.1. Patient Characteristics

Seven centers agreed to participate in the study, including two University Hospitals, three general hospitals, and two for-profit or non-profit private centers (including one cancer center). Out of the initially selected 1028 cases, 816 were eligible for the study, including 413 in 2019 and 403 in 2020. Figure 2 outlines the patient selection process and reasons for ineligibility, most commonly due to the absence of a cancer diagnosis, for the 237 excluded patients.

Patient characteristics in both groups were well distributed, particularly in terms of sex, histology, ECOG PS and treatments administered, including those who underwent surgery. Only two clinical characteristics differed: (1) the 2020 cohort was significantly older (average age 67.6 vs. 65.6 years, *p* = 0.006), with more patients over 70 years old (*p* = 0.002); (2) the 2020 cohort had a higher percentage of non-smoking patients (15% vs. 9.6%). The rates of incidental findings were not higher in the 2020 cohort, and the revealing symptoms were distributed in the same proportions among symptomatic patients. Furthermore, the lockdown period did not appear to impact the treatment center, the specialist type or the diagnostic techniques used. All these results are presented in Table 1.

Moreover, the number of reported cases did not differ between the periods, notably with no significant decrease in diagnosed cases during the lockdown period compared to the same calendar period in 2019 (Figure 3).

### 3.2. Delays for Patient Management

The various delays and their statistical comparisons are outlined in Table 2. Regarding delay 1, no difference was observed between the two populations (HR = 0.89 [0.75–1.04], *p* = 0.15). However, the subgroup analysis (Figure 4) suggested that, while this delay was not modified based on ECOG PS, sex or revealing symptoms, the delay until imaging appeared longer for patients ≥70 years old (*p* = 0.016) and non-smoking patients (*p* = 0.04). On the other hand, the comparison of delay 6 revealed significant differences between the two cohorts (Table 2). This delay appeared shorter in 2020, with an average of 12 days fewer than the 2019 cohort (HR = 1.28 [1.1–1.49], *p* = 0.001). This difference was explained by faster access to specialist consultation (delay 2: HR = 1.18 [1.01–1.38], *p* = 0.048), the diagnostic technique (delay 3: HR = 1.21 [1.03–1.41], *p* = 0.022) and treatment once decided (delay 5: HR = 1.27 [1.09–1.47], *p* = 0.002).

The subpopulations analysis (Figure 5) indicates that the shortening of delay 6 mainly benefited individuals under 70 years old (HR = 1.38, [1.14–1.66], *p* = 0.001), patients with adenocarcinoma (HR = 1.37 [1.11–1.68], *p* = 0.04) or squamous-cell carcinoma (HR = 1.37 [0.99–1.88], *p* = 0.05) and patients treated in general hospitals (HR = 1.42 [1.14–1.76], *p* = 0.001). There was a significant reduction in access delay to surgery for the 2020 cohort (HR= 1.67, [1.19–2.35], *p* = 0.003), although this advantage was also evident for patients receiving systemic treatment (HR = 1.26, [1.05–1.52], *p* = 0.01). However, no statistical difference was observed regarding delay 6 for patients with small-cell neuroendocrine carcinoma or those receiving radiochemotherapy.

### 3.3. Survival Results

Patients in 2020 exhibited significantly higher overall survival compared with the patients in the 2019 cohort (median survival of 17.6 months [14.0–20.7] vs. 13.8 months [11.4–16.0]; HR = 0.80 [0.68–0.95], *p* = 0.01, Figure 6).

Clinical factors statistically associated with improved survival in 2020 were absence of smoking (HR = 0.55, [0.32–0.94], *p* = 0.03), age < 70 years (HR = 0.72 [0.58–0.89], *p* = 0.003), squamous-cell carcinomas (HR = 0.67 [0.48–0.95], *p* = 0.03), ECOG PS 0–1 (HR = 0.75 [0.59–0.94], *p* = 0.01) and 2 (HR = 0.67 [0.48–0.95], *p* = 0.02), Stage IV (HR= 0.75, [0.62–0.9], *p* = 0.003), presence of an EGFR mutation (HR = 0.37 [0.17–0.81], *p* = 0.01) and systemic treatment administration (HR = 0.8 [0.65–0.98], *p* = 0.03).

However, no prognostic differences were observed for patients who underwent surgery or those aged ≥70, patients with ECOG PS 3–4 or those with small-cell neuroendocrine carcinoma (Figure 7). The progression-free survival evaluated in patients receiving treatment was also significantly improved in 2020: 9.2 months [8.3–10.7] vs. 7.5 months [6.8–8.5] (HR = 0.78 [0.66–0.93], *p* = 0.044, Figure 8).

In a multivariate Cox model analysis conducted on the entire study population, being part of the 2020 cohort constituted a favorable independent prognostic factor (HR = 0.71 [0.5–0.84], *p* < 0.001), alongside age < 70 years, female sex, non-smoker status (vs. current or former smokers), Stage I-II vs. III and IV and ECOG PS 0–1 vs. 2 and 3–4 (Table 3).

## 4. Discussion

The hypothesis that the first lockdown implemented during the initial COVID-19 wave may have led to delayed care and potentially impacted the long-term prognosis of cancer patients without COVID-19 is a topic that is still widely debated [18] but not fully resolved. In 2020, a modeling study based on the English national electronic health records suggested a negative impact related to diagnostic delays for four types of solid tumors, including 29,305 LCs [9]. Another study indicated that, among all solid or hematological tumors, LC was the one most strongly correlated with initial consultation, access to imaging and diagnostic delays [19]. However, not all publications had the same conclusions. In Italy, where the first COVID wave had dramatic consequences, paradoxically, no reduction in times from the first symptoms to the treatment was observed [20]. These discrepancies among studies were likely attributed to the vast heterogeneity of the studied populations, the diversity of the types of healthcare centers involved (with varying degrees of engagement in COVID-19 response) and geographic specificities.

Outside of any pandemic context, most prior studies that evaluated the timing of treatment on survival suggested that later treatment could negatively affect prognosis of LC patients [21,22], especially patients with early-stage diagnosis who were waiting for surgery [23,24,25]. However, during the first COVID-19 wave, extended delay to treatment was not necessarily associated with worse overall survival because the patients may have been selected at diagnosis according to their prognostic criteria. For instance, a retrospective study, which selected cases from electronic records, compared different indicators between a pre-COVID-19 cohort (2018–2019) and a 2020 cohort, including the first two COVID-19 waves. While a significant drop in case numbers was noted during the first wave (decrease of 32%), the initial Tumor Node Metastasis stage, the delay to treatment initiation and survival remained unchanged [26]. Likewise, a national study performed on 275,590 stage III-IV LC patients failed to demonstrate a correlation between extended treatment delay from diagnosis and decreased survival compared to prompt treatment [27].

Our study stands out from previously published studies due to its greater robustness on several points. It relied on a systematic medical review of each included case, enabling a near-exhaustive collection of patients’ clinical characteristics and, more importantly, key dates in their management pathways, which cannot be provided by registry studies. The CCFS in which every oncology practitioner commits to report each new case in his charge can be considered a reliable database with little exposure to selection bias. Different types of healthcare centers, whether public or private, participated. Finally, to our knowledge, this study is the only one published with a median follow-up for overall survival exceeding 2 years.

The comparison between the cohorts indicated that the first lockdown and the accompanying health restrictions had only a minor influence on patient management and outcomes. In terms of clinical characteristics, the initial concern was that new patients in 2020 might have received a later-stage LC diagnosis compared with pre-pandemic times, due to longer delays. However, this hypothesis was not validated by the comparison of ECOG PS and the different stages, which were identical between the two groups. It was also suggested that a certain number of cases might have been diagnosed during thoracic imaging exams performed for suspected COVID-19 infection. The similarity in the proportion of early stages and incidental discoveries suggested that any potential occurrence of such a phenomenon was probably minimal. General practices were not modified by health restrictions, since patient distributions according to type of center, the type specialists providing care or the examination used for diagnosis were the same for each year. Regarding treatments, no differences were observed in proportions based on the type of treatment delivered, even though the conversion from surgical procedures to stereotactic radiotherapy during the pandemic has been reported elsewhere [28]. Only the age of patients and the proportion of non-smoking patients were higher in 2020, and we believe that this is an epidemiological trend specific to LC, unrelated to the pandemic, as demonstrated by certain studies [29]. Interestingly, we did not find a significant drop in the number of cases diagnosed during this first wave, which is contrary to what was observed in other cohorts. We believe that this result is due to several factors: first, the pandemic wave was less strong in the PL region and, therefore, the patients were managed within the usual time frame; second, some centers did not take any COVID-19 patients, meaning their organization of care was not disrupted.

In order to evaluate whether certain phases of patient management were prolonged due to health measures, seven delays (including four based on the National Institute of Cancer (INCa criteria)) were defined using seven different dates, ranging from the first symptom to the first treatment. Regarding the period preceding the disease discovery, there was concern about underestimating symptoms, especially in patients at risk of severe COVID-19, who needed to be kept away from medical facilities where the infection was prevalent. However, the comparison between the two cohorts regarding the delays between the first symptoms and the first imaging failed to show any difference; thus, our study did not support the hypothesis that LC diagnosis was neglected during this period. However, there was one exception: the subgroup analysis showed that this delay was statistically longer in patients over 70 years old, suggesting that elderly patients may have been kept in lockdown longer, possibly leading to diagnostic delays.

The comparison of delay 6, representing the entire pre-therapeutic medical management, including the diagnostic phase and the delay for the first treatment, led to unexpected results. Indeed, the three principal dates during which practitioners could potentially have influenced the process—namely, the date of the first consultation, the diagnostic date and the date of the first treatment—were significantly advanced in 2020 compared with 2019. This suggested that the cancellation or rescheduling of consultations, technical procedures or hospitalization sessions scheduled for other non-urgent conditions might, paradoxically, have been beneficial for new LC cases.

The most surprising finding was in surgical cases, where the initial concern was the rescheduling of surgical procedures due to operating room closures. Nevertheless, in our study, the delay between the first imaging and the surgery was shortened by an average of 22 days in 2020 (HR = 1.67 [1.19–2.35], *p* = 0.003), contradicting this hypothesis. Access to systemic treatment of any kind was also shortened during the pandemic period. Patient management was accelerated for every clinical factor, except for small-cell LC, which requires urgent treatment, and for patients treated with radiotherapy, whose organization is more complex than other therapies. For these two situations, the waiting time for treatment was incompressible. Of note, the decrease in delay was more important in general hospitals, likely because some of them were less involved in the fight against COVID-19 compared to university hospitals, for example.

To our knowledge, the prognostic impact of the first lockdown has never been evaluated after a sufficient follow-up period [26], making this public health issue unresolved to date. The multivariate survival analysis demonstrated the better prognosis of patients in the 2020 cohort, independently of other prognostic covariates such as age, stage, ECOG PS, sex and smoking status. The improved prognosis of Stage IV patients, those who received systemic treatment or those with adenocarcinoma or squamous-cell carcinoma, suggests that the broadening of indications for immunotherapy or access to new targeted therapies contributed to this outcome. Indeed, the chemo-immunotherapy was authorized for prescription as early as October 2019 for adenocarcinoma and April 2020 for squamous-cell carcinoma, while certain targeted therapies were granted early access (sotorasib for KRas G12C mutations in the second-line since June 2021) or were routinely prescribed (osimertinib as first-line therapy since October 2020). On the other hand, subgroup analyses showed that the prognosis improvement was not systematically correlated with a shortening of delays (as seen in cases like squamous-cell carcinomas or surgical cases), suggesting a minor impact of the lockdown on patients’ outcomes.

However, the ARTEMISIA study bears certain limitations that may reduce its scientific scope. Firstly, the size of the cohort may be considered small considering the targeted goal. However, we chose to limit the inclusion periods to 6 months each, as our aim was to assess the consequences of the first lockdown on patient management. Moreover, new restrictions were announced after the summer of 2020, which could have caused confusion in interpreting the results. Nonetheless, we acknowledge that some patients may have been managed later, i.e., after the inclusion deadline. Moreover, the PL region in western France, where our study was conducted, did not experience the most severe consequences of the pandemic, as the reported cases and death numbers were below the national average (https://www.insee.fr/fr/statistiques/4500439 (accessed on 26 June 2020).

Several centers, particularly private ones that predominantly handle surgery and radiotherapy, were unable to participate in the study due to resource constraints, which could have introduced a selection bias. Lastly, we did not document COVID-19 infection cases to evaluate their prognostic impact. We considered that collecting COVID-19 diagnoses from the hospital medical records would not be reliable, as most reported cases were based on external samples without systematic transcription into patient records. Additionally, infections were rare at that time, and their prognosis among LC patients was already known and published [26,30,31].

## 5. Conclusions

The ARTEMISIA study suggested that the health restrictions implemented during the first wave of the COVID-19 pandemic did not impact the management and prognosis of LC patients. However, the aim of this work was not to refute what other studies have demonstrated in other locations, but only to present a realistic picture of the pandemic and its consequences for patients treated in a European region that was less affected by the pandemic and where protective measures were strictly adhered to. In more general terms, it illustrated the responsiveness and adaptability of a national healthcare system in thoracic oncology.

## Figures and Tables

**Figure 1 cancers-15-05729-f001:**
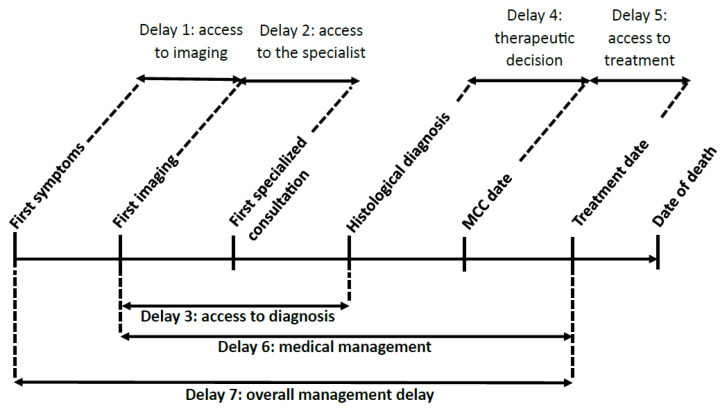
Definitions of management delays. MCC: multidisciplinary cancer conference.

**Figure 2 cancers-15-05729-f002:**
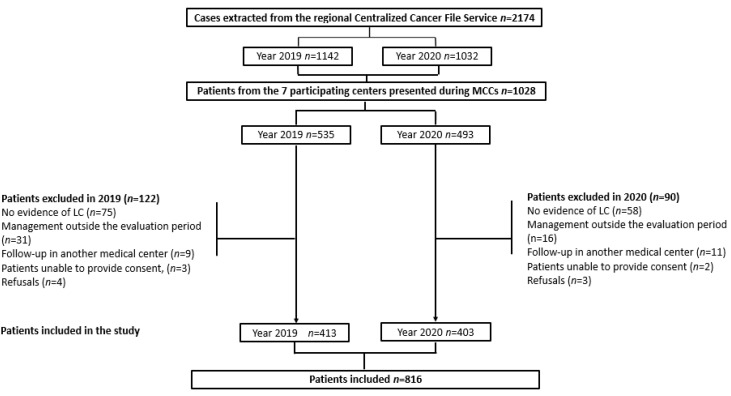
Patient selection. MCCs: multidisciplinary cancer conferences, LC: lung cancer.

**Figure 3 cancers-15-05729-f003:**
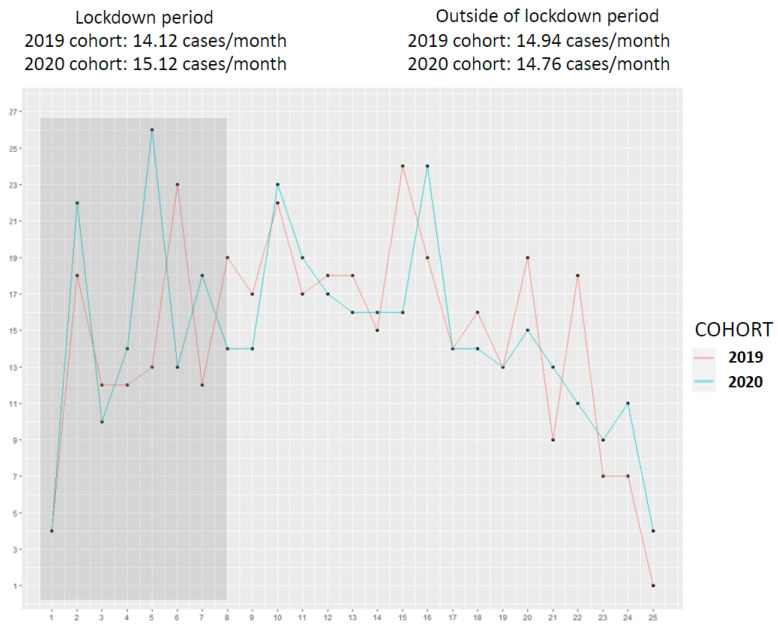
Moving average of monthly lung cancer case diagnoses within the two cohorts.

**Figure 4 cancers-15-05729-f004:**
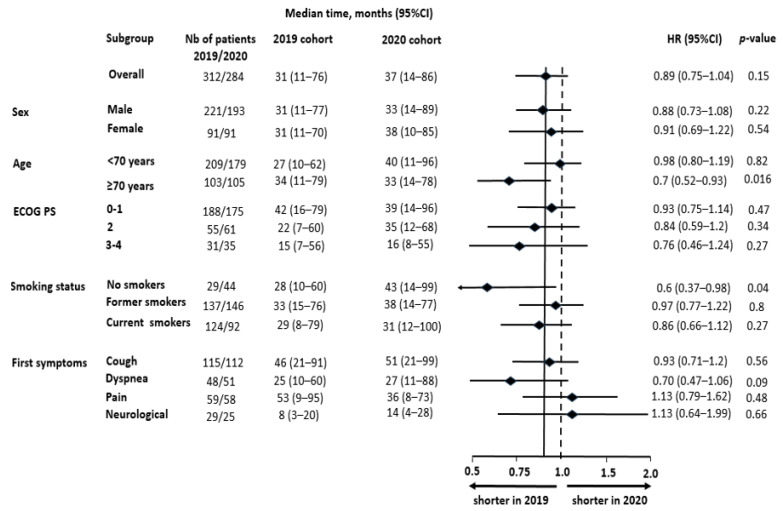
Subgroup analysis of delay 1, defined as the time frame from first symptoms to first imaging. Non-smokers: patients who smoked fewer than 100 cigarettes in their lifetime, Former smokers: patients declaring smoking cessation since at least 1 year prior to diagnosis, ECOG PS: Eastern Cooperative Oncology Group performance status, HR: hazard ratio, CI: confidence interval.

**Figure 5 cancers-15-05729-f005:**
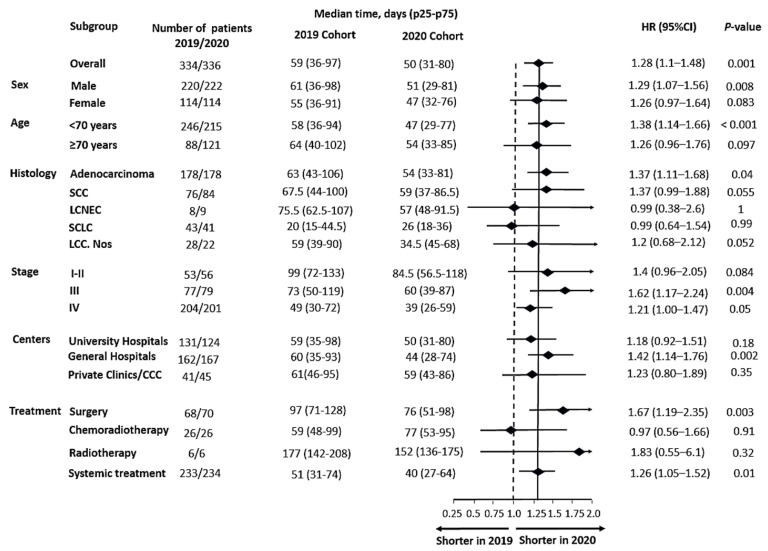
Subgroup analysis of delay 6, defined as the time frame from the first suspicion of lung cancer (date of first imaging) to the date of first treatment. CCC: cancer comprehensive center, SCC: squamous-cell carcinoma, LCNEC: large-cell neuroendocrine carcinoma, SCLC: small-cell lung cancer, LCC Nos: large-cell carcinoma, not otherwise specified, HR: hazard ratio, CI: confidence interval.

**Figure 6 cancers-15-05729-f006:**
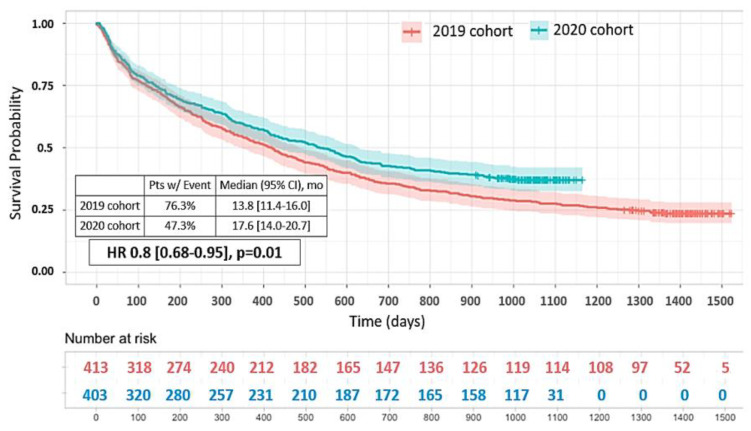
Overall survival. Tick marks indicate censored data.

**Figure 7 cancers-15-05729-f007:**
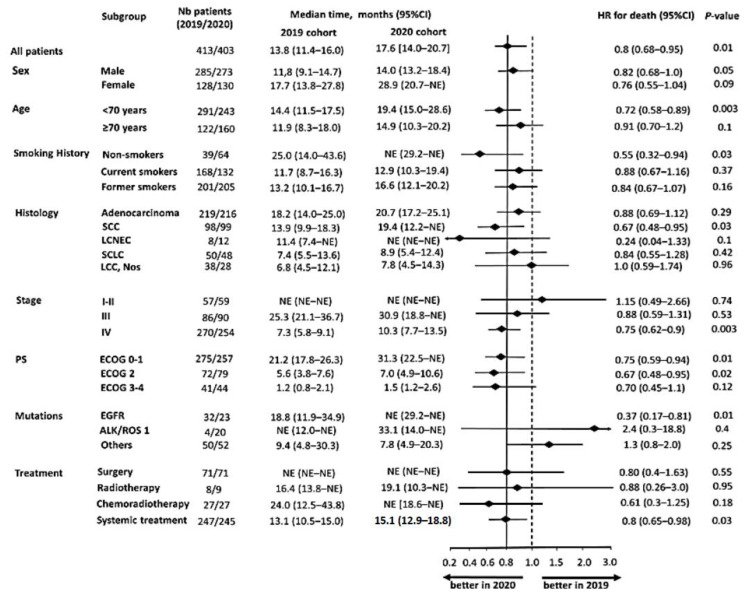
Subgroup analysis for overall survival.

**Figure 8 cancers-15-05729-f008:**
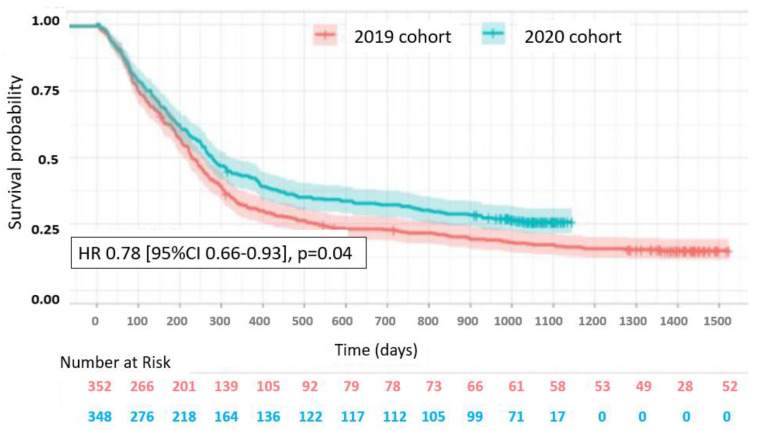
Progression-free survival in treated patients. Tick marks indicate censored data.

**Table 1 cancers-15-05729-t001:** Descriptions of the 2019 and 2020 cohorts.

			Population 2019 *n* = 413		Population 2020 *n* = 403		
Sex		*n*	413		403		0.697
	Male	*n* (%)	285	(69.0)	273	(67.7)	
	Female	*n* (%)	128	(31.0)	130	(32.3)	
Age		*n*	413		403		0.006
	Mean age	(sd)	65.6 ans	(10.5)	67.6 ans	(10.3)	
Age ≥ or <70 years old		*n*	413		403		0.002
	≥70 years	*n* (%)	122	(29.5)	160	(39.7)	
	<70 years	*n* (%)	291	(70.5)	243	(60.3)	
Smoking status		*n*	408		401		
	Non-smoker	*n* (%)	39	(9.6)	64	(15.9)	0.006
	Former smoker	*n* (%)	201	(49.3)	205	(51.2)	
	Active smoker	*n* (%)	168	(41.1)	132	(32.9)	
Disease discovery		*n*	413		403		
	Symptomatic patients	*n* (%)	329	(79.7)	328	(81.4)	0.533
	Incidental finding	*n* (%)	84	(20.3)	75	(18.6)	
Type of center		*n*	413		403		0.923
	University Hospital	*n* (%)	159	(38.5)	151	(37.5)	
	General Hospitals	*n* (%)	201	(48.7)	197	(48.9)	
	Private Clinics or CCC	*n* (%)	53	(12.8)	55	(13.6)	
Symptoms		*n*	314		315		0.444
	Cough	*n* (%)	135	(43.0)	128	(40.6)	
	Dyspnea	*n* (%)	52	(16.6)	61	(19.4)	
	Pain	*n* (%)	66	(21.0)	68	(21.6)	
	Hemoptysis	*n* (%)	18	(5.7)	10	(3.2)	
	Neurological	*n* (%)	30	(9.6)	28	(8.9)	
	Others	*n* (%)	13	(4.1)	20	(6.3)	
Histology		*n*	413		403		0.688
	Adenocarcinoma	*n* (%)	219	(53.0)	216	(53.6)	
	SCC	*n* (%)	98	(23.7)	99	(24.6)	
	SCLC	*n* (%)	50	(12.1)	48	(11.9)	
	LCNEC	*n* (%)	8	(1.9)	12	(3.0)	
	LCC, Nos	*n* (%)	38	(9.2)	28	(6.9)	
ECOG		*n*	388		380		0.62
	0–1	*n* (%)	275	(70.8)	257	(67.6)	
	2	*n* (%)	72	(18.6)	79	(20.8)	
	3–4	*n* (%)	41	(10.6)	44	(11.6)	
Stage		*n*	413		403		0.782
	I–II	*n* (%)	57	(13.8)	59	(14.6)	
	III	*n* (%)	86	(20.8)	90	(22.3)	
	IV	*n* (%)	270	(65.4)	254	(63.0)	
Specialist type		*n*	412		403		0.005
	Pulmonologist	*n* (%)	368	(89.3)	351	(87.1)	
	Oncologist/radiotherapist	*n* (%)	14	(3.4)	6	(1.5)	
	Surgeon	*n* (%)	17	(4.1)	38	(10.8)	
	Others	*n* (%)	13	(3.5)	8	(2.3)	
Diagnostic examination		*n*	411		382		0.951
	Bronchoscopic Fibroscopy	*n* (%)	174	(42.3)	157	(41.1)	
	Image-guided biopsy	*n* (%)	120	(29.2)	125	(32.7)	
	Serous puncture		23	(5.6)	23	(6.0)	
	EBUS		24	(5.8)	26	(6.8)	
	Exploratory surgery	*n* (%)	32	(7.8)	30	(7.9)	
	Curative surgery	*n* (%)	38	(9.2)	41	(10.7)	
Molecular Alterations		*n*	413		403		0.003
	Absence		327	(74)	328	(70.3)	
	EGFR	*n* (%)	32	(9.8)	23	(7.8)	
	ALK/ROS1	*n* (%)	4	(0.9)	20	(6.1)	
	Others	*n* (%)	50	(15.3)	52	(15.8)	
PD-L1 TPS		*n*	296		282		0.638
	<1%	*n* (%)	121	(40.9)	122	(43.3)	
	49–50%	*n* (%)	84	(28.4)	70	(24.8)	
	≥50%	*n* (%)	91	(30.7)	90	(31.9)	
Treatments		*n*	413		403		0.995
	Surgery		71	(17.2)	71	(17.6)	
	Curative radiotherapy		8	(1.9)	9	(2.2)	
	Radiochemotherapy		27	(6.5)	27	(6.7)	
	Systemic treatment		247	(59.8)	245	(60.8)	
	Upfront supportive care		60	(14.5)	51	(12.6)	

Non-smokers: patients who smoked fewer than 100 cigarettes in their lifetime, Former smokers: patients declaring smoking cessation since at least 1 year prior to diagnosis, SCC: squamous-cell carcinoma, LCNEC: large-cell neuroendocrine carcinoma, SCLC: small-cell lung cancer, LCC Nos: large-cell carcinoma, not otherwise specified, ECOG: Eastern Cooperative Oncology Group, PS: performance Status, PS 0: fully active, PS 1: restricted in heavy physical work, PS 2: up and about more than half-day, PS 3: in bed or sitting in a chair more than half-day, PS4: in bed or in a chair all the time, HR: hazard ratio, CI: confidence interval, NE: not evaluated, EBUS: endobronchial ultrasound, EGFR: epidermal growth factor, ALK: anaplastic lymphoma kinase, ROS1: ROS proto-oncogene 1, PD-L1: programmed death ligand 1, TPS: tumor proportion score.

**Table 2 cancers-15-05729-t002:** Management delays.

	Cohort	Mean Days ± Standard Deviation	Median Days (25th–75th Percentiles)	HR [CI 95%]	*p*
Delay 1: access to imaging	2019	53.8 ± 69.3	31 (11–76.2)	0.89 [0.75–1.04]	0.15
	2020	61.7 ± 80.4	37 (14–85.8)		
Delay 2: access to the specialist	2019	16.9 ± 24.5	8 (2–21)	1.18 [1.01–1.38]	0.048
	2020	14.2 ± 28.0	6 (1–16)		
Delay 3: access to diagnosis	2019	38.3 ± 37.5	25 (11–53)	1.21 [1.03–1.41]	0.022
	2020	33.2 ± 40.4	20 (10–39)		
Delay 4: therapeutic decision	2019	8.7 ± 34.0	7 (2–15)	1.02 [0.89–1.17]	0.78
	2020	7.7 ± 28.2	7 (2–14)		
Delay 5: access to treatment	2019	23.5 ± 35.6	17 (7–33)	1.27 [1.09–1.47]	0.002
	2020	18.8 ± 18.1	14 (6–27)		
Delay 6: medical management	2019	72.0 ± 52.2	59.5 (36–97)	1.28 [1.1–1.49]	0.001
	2020	60.3 ± 41	50 (30.8–80)		
Delay 7: overall management delay	2019	120 ± 84.3	100 (67.5–154)	1.00 [0.84–119]	0.97
	2020	119 ± 96.5	97 (56.2–158)		

HR: hazard ratio, CI: confidence interval.

**Table 3 cancers-15-05729-t003:** Multivariate analysis of overall survival.

Characteristics	HR	95% CI	*p*-Value
Age ≥ 70 vs. <70	1.29	1.06–1.57	0.01
Female vs. male	0.64	0.52–0.78	<0.001
Smoking status			
Non-smoker	-		
Current smoker	1.73	1.23–2.45	0.002
Former smoker	1.74	1.26–2.41	0.001
Stage			
I–II	-		
III	3.73	2.28–6.1	<0.001
IV	7.75	4.85–12.39	<0.001
PS			
0–1	-		
2	2.3	1.84–2.84	<0.001
3–4	4.49	3.45–5.85	<0.001
2020 vs. 2019	0.71	0.59–0.84	<0.001

## Data Availability

The datasets used or analyzed in the current study are available from the corresponding author upon reasonable request.

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
