# Peer review of "A Comparative Multicenter Cohort Study Evaluating the Long-Term Influence of the Strict Lockdown during the First COVID-19 Wave on Lung Cancer Patients (ARTEMISIA Trial)"

_cancers, 2023, doi:10.3390/cancers15245729_

Round 1
Reviewer 1 Report
Comments and Suggestions for Authors
Perhaps too short a time period of analysis to compare the two groups.
It has been published in the lung cancer medical literature that small but statistically significant differences are described between individual hospitals within one country regarding treatment and accessibility for patients in individual regions of a large country. Therefore, it would be appropriate to make further comparisons only for each individual hospital in one region and therefore an important observation for a longer period of time.
Otherwise, this research is valuable for clinical practice in the future as well.
Author Response
You are right, there are differences in our region which are already well known: private centers tend to provide radiotherapy and surgery, while public centers participate in the diagnostic phase and systemic treatment. If we had made comparisons, we would certainly have found differences, but we think that is another debate, this analysis would not have provided tangible elements on the impact of COVID on the outcome of patients. Note that we compared 3 types of establishments (private, general public and university hospital) in terms of treatment times
Reviewer 2 Report
Comments and Suggestions for Authors
This paper is an analysis of the impact of COVID on lung cancer diagnosis interval and the potential impact on survival. The comparison of two cohorts pre and COVID epidemic patients is useful and potentially relevant. Howevre, I think there are several points that should be considered by the authors:
1- The cohorts are hospital based, not population based. I think it is needed to know which percenatge of all cases diagnosed in the region are included in the study and some demographic information of these patients not included to assess the possible selection bias.
2. It is quite surprising that the number of case are similar in both years. One reported impact of the cOVID was an underdiagnosis of new cases at the initial wave of the epidemia. This point should be discussed by the authors.
This point also could clarify the number of cases, quite reduced considering that the region is of almost 4M inhabitants
3. The literature on COVID is enormous, I wonder if only with 8 references is enough. The review of the literature would take advantage of a more detailed comment on the impact of COVID observed in other studies.
4. Have you tried to include the intervals in the multivariate model to asses its potential power to explain the results in survival?
5- Simple summary: perhaps it would need to include some quantitative results in the text to be a proper summary.
Comments on the Quality of English Language
NO specific comments on language
Author Response
1- For obvious reasons of confidentiality, we can't have the exact number of cases diagnosed in the region and clinical informations about them. On the other hand, we have inquired about the exact number of cases discussed in MCC throughout the region during the 2 study periods. The results with the proportions appear in figure 2. There is indeed a selection bias since the centers not participating in the study (because they don't have a research team) are mainly private centers, whose main activity in thoracic oncology is surgery. This point also appears in the discussion, in the limitations paragraph.
2- You are right, we neglected to discuss this point. The hypothesis to explain this result, which in fact goes against the results of the majority of studies, is that the magnitude of the pandemic wave was weaker than in other places, so that patients could be taken in charge within the usual deadlines. In addition, some centers didn't take any COVID patients and their organiztion was not disrupted. We added these arguments in the discussion.
3- we have expanded the bibliographic references as well as the introduction which you considered insufficient.
4- We of course thought about it but did not find a multivariate statistical method capable of evaluating a delay as a prognostic factor, the Cox method not being usable. Finally, the gain in treatment time is derisory (9 days on median) compared to the gain in survival for the 2020 cohort (more than 3 months) and we think it is unlikely that it could have played any role in the improvement. of the prognosis. We have simply assumed in the discussion what seems to us to be the most obvious: better survival in 2020 due to new treatments.
5- we added some numerical results in the summary but we are limited on the number of words
Reviewer 3 Report
Comments and Suggestions for Authors
This article explored the influence of the strict lockdown on outcomes of lung cancer patients during the first COVID-19 wave. In this study, the authors analyzed two cohorts, in which one was exposed during the lockdown period and the other lived before the pandemic. This study is meaningful and interesting. However, there are several issues regarding this manuscript:
1. How to define the data of first symptoms? Symptoms, such as pain, and cough could be caused by other diseases other than lung cancer.
2. In the subgroup analysis of delay 6, some factors (age<70, adenocarcinoma, general hospital, surgery) contributed to accelerating patient management in the 2020 cohort. The authors should explain the reasons in the discussion.
3. Even if the strict lockdown did not exert a detrimental impact on lung cancer patients, it’s hard to understand why the 2020 cohort has a significantly better OS than the 2019 cohort. In multivariate Cox model analysis, 2020 vs 2019 was also included as an independent prognostic factor. Rapid advances in lung cancer therapy in 2020 or survivor effect? The epidemic might further increase the mortality rate of the 2019 cohort.
4. The decimal digits in your tables should be unformed.
5. Figures are in low resolution and difficult to view.
6. Figure 6 lacks labels and its title is mixed with annotation.
Author Response
1- each case was reread from the medical file. based on the clinical description and imaging examinations, we considered each symptom as disease-related or not. But you are right, it is easier to judge pain or hemoptysis than dyspnea or cough.
2- If the results are significant for young patients or adenocarcinomas, it is certainly because they are the most represented populations (there is no real difference in HR between elderly and non-elderly patients, or between squamous and adenocarcinomas; it is therefore a question of statistical power). except for small cell and radiotherapy (explanations provided in the discussion). We have modified our comment, to be better understood).
3- Even though cases of COVID were severe in lung cancer patients with a high risk of mortality, they were rare at the time precisely thanks to the confinement. The epidemic certainly had little influence on the mortality of this population at the time and, as we said in the discussion, it was probably the new treatments and new treatment strategies, including immunotherapy and targeted therapies which led to a better survival between the 2 cohorts.
5 and 6- we have corrected
Round 2
Reviewer 2 Report
Comments and Suggestions for Authors
Thanks for your reaction to my comments. I have no further comments